# Basal leakage in oscillation: Coupled transcriptional and translational control using feed-forward loops

Ignasius Joanito[1,2¤], Ching-Cher Sanders Yan[1], Jhih-Wei Chu[2,3], Shu-Hsing Wu[4,5], Chao-Ping Hsu[1,5]*

1 Institute of Chemistry, Academia Sinica, Taipei, Taiwan, 2 Bioinformatics Program, Taiwan International Graduate Program, Academia Sinica, Taipei, Taiwan and Institute of Bioinformatics and System Biology, National Chiao Tung University, Hsinchu, Taiwan, 3 Department of Biological Science and Technology, National Chiao Tung University, Hsinchu, Taiwan, 4 Institute of Plant and Microbial Biology, Academia Sinica, Taipei, Taiwan, 5 Genome and Systems Biology Degree Program, National Taiwan University, Taipei, Taiwan

¤ Current address: Computational and Systems Biology, Genome Institute of Singapore, Singapore, Singapore

* cherri@sinica.edu.tw

**Data Availability Statement:** All relevant data are within the manuscript and its Supporting Information files.

## Abstract

The circadian clock is a complex system that plays many important roles in most organisms. Previously, many mathematical models have been used to sharpen our understanding of the *Arabidopsis* clock, which brought to light the roles of each transcriptional and post-translational regulations. However, the presence of both regulations, instead of either transcription or post-translation, raised curiosity of whether the combination of these two regulations is important for the clock's system. In this study, we built a series of simplified oscillators with different regulations to study the importance of post-translational regulation (specifically, 26S proteasome degradation) in the clock system. We found that a simple transcriptional-based oscillator can already generate sustained oscillation, but the oscillation can be easily destroyed in the presence of transcriptional leakage. Coupling post-translational control with transcriptional-based oscillator in a feed-forward loop will greatly improve the robustness of the oscillator in the presence of basal leakage. Using these general models, we were able to replicate the increased variability observed in the E3 ligase mutant for both plant and mammalian clocks. With this insight, we also predict a plausible regulator of several E3 ligase genes in the plant's clock. Thus, our results provide insights into and the plausible importance in coupling transcription and post-translation controls in the clock system.

## Author summary

For circadian clocks, several current models had successfully captured the essential dynamic behavior of the clock system mainly with transcriptional regulation. Previous studies have shown that the 26S proteasome degradation controls are important in maintaining the stability of circadian rhythms. However, how the loss-of-function or over-

**Funding:** We acknowledge support from Academia Sinica through the Investigator Award (AS-IA-106-M01), the Ministry of Sciences and Technology of Taiwan (grant no. MOST 105-2113-M-001 -009 -MY4), and a joint research grant from National Taiwan University and Academia Sinica (NTU-AS-106R104517) for support of this research work. The funders had no role in study design, data collection and analysis, decision to publish, or preparation of the manuscript.

**Competing interests:** The authors have declared that no competing interests exist.

expression mutant of this targeted degradations lead to unstable oscillation is still unclear. In this work, we investigate the importance of coupled transcriptional and post-translational feedback loop in the circadian oscillator. With general models our study indicate that the unstable behavior of degradation mutants could be caused by the increase in the basal level of the clock genes. We found that coupling a non-linear degradation control into this transcriptional based oscillator using feed-forward loop improves the robustness of the oscillator. Using this finding, we further predict some plausible regulators of Arabidopsis's E3 ligase protein such as COP1 and SINAT5. Hence, our results provide insights on the importance of coupling transcription and post-translation controls in the clock system.

## Introduction

The circadian clock is an endogenous time-keeping mechanism in cells that anticipates daily changes in the environment [1–4]. It controls the daily rhythm of many biological processes [5–7] and disruption of the clock has been associated with many disadvantageous traits [8–11]. Like many eukaryotes, in the *Arabidopsis* clock, the core oscillator is governed by coupled transcription and translation feedback loops (TTFL) [12]. The transcriptional circuit consists of several important genes such as *CIRCADIAN CLOCK-ASSOCIATED1* (*CCA1*), *LATE ELONGATED HYPOCOTYL (LHY)*, *TIMING OF CAB EXPRESSION 1 (TOC1)*, *PSEUDO-RESPONSE REGULATOR 9 (PRR9)*, *PRR7*, and *PRR5* [12]. Experimentally, *CCA1* and *LHY* genes were found to repress the expression of *TOC1*, *PRR9*, *PRR7*, and *PRR5* [13–15], whereas in turn, all of them repressed *CCA1* and *LHY* expression [16–19]. Furthermore, *TOC1* and *PRR5* can repress *PRR9* and *PRR7* expressions [17,20]. Together, they formed a 3-node loop of negative regulation, the repressilator [21]. The core repressilator motif is coupled with positive feedback loops, leading to several interesting properties [22–24].

For the post-translational process, several regulations, such as protein–protein interaction [25,26], subnuclear localization [27], phosphorylation [28], and 26S proteasome degradation [29,30], have been reported in the past decades. Among these many regulations, the 26S proteasome degradation pathway, also known as ubiquitin-proteasome system (UPS), attracts our attention. UPS involves in almost all aspects of plant life cycle, such as root elongation, light response, flowering time, seed development and also biotic and abiotic stress (for comprehensive review, see [31,32]). Experimentally, almost all important clock genes, including *CCA1* [33], *LHY* [34,35], *TOC1* [29], *PRR9* [36], *PRR7* [37], *PRR5* [38], *GIGANTEA* (*GI*) [39], *EARLY FLOWERING 3* (*ELF3*) and *CONSTITUTIVE PHOTOMORPHOGENIC* 1 (*COP1*) [40], were found degraded through the 26S proteasome degradation pathway. These observations imply that the 26s proteasome degradation pathway is important in the *Arabidopsis* clock system.

Remarkably, such degradation control is also found in other clock systems such as mammals, flies, and fungi [2–4,41]. In the mammalian clock, two important clock components, cryptochrome (*CRY*) and period (*PER*), are also degraded through the 26S proteasome degradation pathway. *CRY* protein is targeted for proteasomal degradation by two different F-box proteins, *Fbxl3* and *Fbxl21* [42–46]. However, mutations in these two E3 ligase proteins did not alter the behavior of circadian rhythms markedly as compared with phenotypes that were caused by mutations in other clock genes [47]. A recent study showed that alteration of *β-Trcp1* and *β-Trcp2* proteins, other F-box proteins that will trigger the degradation of *PER* protein, severely altered the clock's function. In the *β-Trcp* double-mutant mice, the oscillation

of many clock genes are highly unstable, as indicated by the greatly increased variability in circadian rhythm [47].

The striking increase in variability of E3 ligase mutant plant has also been seen in the *Arabidopsis* clock. Previously, an F-box protein, *ZEITLUPE (ZTL)*, was found involved in target degradation of both *PRR5* and *TOC1* [29,38]. Somers *et al.* showed that mutation of *ZTL* protein would greatly increase the variability of both amplitude and period of circadian oscillations [30]. To the best of our knowledge, why this mutation can lead to high variability in the plant's clock is still unknown. For the mammalian clock, D'Alessandro *et al.* proposed that such unstable circadian rhythms in *β-Trcp* mutant mice may come from loss of nonlinear degradation of *PER* protein [47]. However, why such loss of nonlinear degradation can lead to unstable behavior is still not clear.

Hence, in this study, we developed a series of simplified models to study the importance of targeted degradation in the clock system. Our results showed that basal leakage in the transcription of clock genes leads to unstable behavior of the clock system, which can be stabilized by the degradation control. Here, we showed that basal leakage could easily reduce the robustness of an oscillator, especially for a pure transcriptional controlled oscillator. However, combining transcriptional and post-translational controls can greatly improve the robustness of the oscillator by providing another layer of regulation such that the system can still push the protein level back, despite leakage in the mRNA level. Moreover, we also found that coupling E3 ligase using a coherent feed-forward loop (CFFL) can give better control to the basal leakage as compared with other network motifs. Using these general models, we have successfully replicated the observed experimental results of the *ZTL* and *β-Trcp* mutant conditions and also predict plausible regulators of several E3 ligases in *Arabidopsis*. Therefore, our results provide plausible importance in coupling transcription and post-translation controls in the clock system.

## Results

### Transcription-based oscillators have lower noise but are susceptible to transcriptional leakage

In this study, we built a series of simple oscillator systems based on the repressilator, with different regulation controls (Fig 1A). A repressilator is a circular three-inhibitor feedback loop that was originally constructed as a synthetic circuit and capable of generating oscillation in *Escherichia coli* [21]. However, in recent years, repressilators can also be found in many oscillating systems such as *Arabidopsis* [48] and mammalian [49] circadian clocks. The system has also been widely used to study many interesting properties of an oscillator such as tunability [50] and switchability [51,52]. Therefore, we used a repressilator to study the effect of different regulation controls in the oscillator, including transcription (M1), post-translation (M2), transcription with positive auto-regulation (M3), post-translation with positive auto-regulation (M4), and combined transcription and post-translational control (M5) (Fig 1A). Here, we randomly selected the parameter sets, from a uniform distribution in their log scale, for each model such that their deterministic dynamics would oscillate and then performed Gillespie's stochastic simulation [53,54]. The robustness of the oscillation was measured by estimating the normalized decay rate of the autocorrelation function (Fig 1B, Methods).

For a simple transcriptional feedback model, M1, among 1,000 parameter sets that can oscillate under deterministic propagation, 905 (90.5%) yielded sustained oscillation under a noisy condition, and their normalized decay rates have a median value of 1.04 (Fig 1C). For the post-translational feedback model, M2, oscillatory parameter sets under a noisy condition were found at a lower rate: 344 of 1,000 (34.4%), with much worse median value of 2.16 in

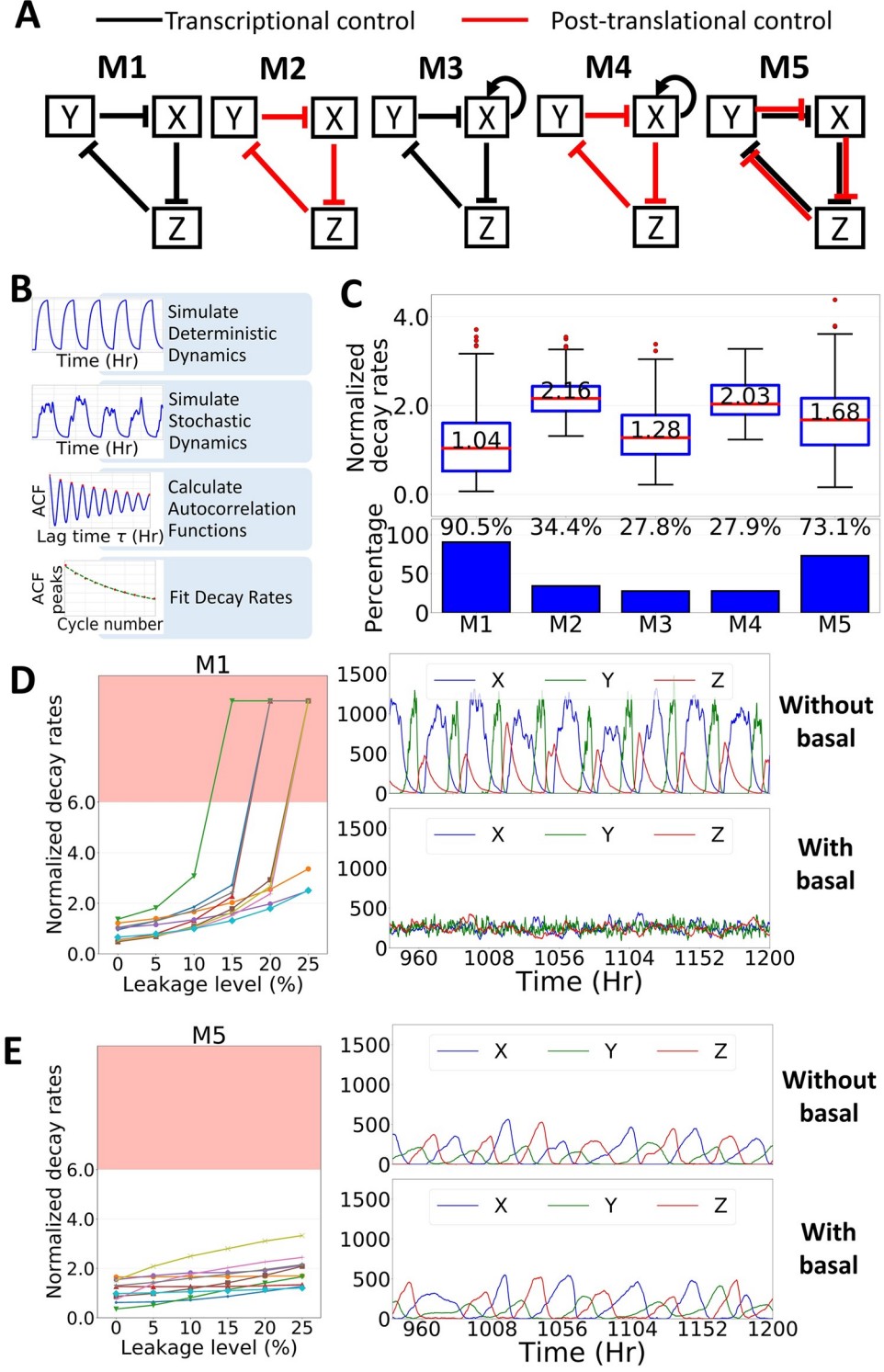

**Fig 1. Combining transcriptional and post-translational controls improves the robustness of an oscillator.** (A) Schematic representation of the tested models. (B) General workflow used in this study. (C, Upper panel) Box plot representing normalized decay rates of each tested model for 1000 randomly select parameter sets. Red lines indicate the median, and box edges indicate the 25th (Q1) and 75th (Q3) percentiles. Whiskers are plotted at 1.5*(Q3-Q1). (C, lower panel) Percentage of parameter sets that showed sustained oscillation under stochastic simulations. (D, E, left) the normalized decay rates of 10 randomly chosen parameter sets of M1 (D) or M5 (E) under different leakage levels. Each line represents one parameter set. The red-shaded region of the plot indicates non-oscillating results. (D, E, right) Time trace trajectory of one randomly chosen parameter sets of M1 (D) or M5 (E).

decay rates (Fig 1C). This result may occur because, in M2, the noise from uncontrolled mRNA was propagated to the protein level, which easily altered the phase and period of the oscillation (S1 Fig). Furthermore, adding an auto-positive feedback on a transcriptional-based oscillator (M3) or post-translational–based oscillator (M4) yielded oscillations at an even lower rate: 278 of 1,000 parameter sets (27.8%) with median value of 1.28 for M3, and 279 of 1,000 parameter sets (27.9%) with median value of 2.03 for M4. Thus, our results showed that a simple transcriptional-based oscillator is more robust than a post-translational-based oscillator.

Next, we tested the oscillators for transcriptional leakage, which commonly occurs in cells. Many studies have shown that promoters are actually leaky [55–58]. Yanai *et al.* in 2006, even suggested that the selection against "unnecessary" transcription is low and hypothesized that leakiness of the promoters may be evolutionarily neutral [59,60]. Hence, we added an additional term to the gene production to represent this leakage in our models, and re-tested the performance of M1 (see Methods for more detailed information). Here we found that adding transcriptional leakage greatly reduced the robustness of model M1 in generating oscillation (Fig 1D). Actually, the system was very sensitive such that adding 5% leakage reduced the number of oscillating parameters sets from 90.5% to 7.3% (S2A Fig). Furthermore, adding a positive feedback did not improve the performance of the transcriptional-based oscillator (S2B Fig). Therefore, these results suggested that although a simple transcriptional-based oscillator can already generate good oscillation, it is prone to transcriptional leakage that may be present in cells.

## Combining transcription and post-translation controls improves the robustness of the oscillator

We proposed to combine both a transcriptional and post-translational control for a more robust oscillator. The basic idea is that although the post-translation-based model, M2, has much worse oscillation in a stochastic condition (higher normalized decay rate) compared to M1, it will not have any leakage problem, since the transcription in M2 is not regulated. Instead, post-translational control may provide another layer of regulation for the repression leakage that occurred at the transcription level. To test this idea, we built a simple model, M5, that combined both M1 and M2 and performed a similar test as described above (Fig 1B). We found that 731 out of 1,000 parameter sets (73.1%) yielded sustained oscillation under a noisy condition, with a normalized decay rate median value of 1.68 (Fig 1C). Although the median value was still higher than M1, unlike M2, model M5 had a wider normalized decay rates distribution. It implied that M5 could still achieve robust oscillation under correct combination of parameters. Moreover, we also found that M5 was much more robust as compared with M1 under the basal leakage condition, such that 38.1% of parameter sets were still oscillating under 5% basal leakage (Fig 1E and S2C Fig). In addition, we also found that 16.3% of the parameter sets in model M5 were still able to oscillate even at 25% leakage level (compared to only 0.3% of parameter sets in model M1) (S2A and S2C Fig). These results imply that model M5 may have a unique property to handle the basal leakage.

Adding a post-translational control in the transcriptional-based oscillator may help the system regulate the leakage in the transcriptional repression, such that it can still push the protein level back despite the leakage in the mRNA level. To demonstrate this idea, we showed that although we could still observe obvious shifting in the mRNA distribution for both models when we added a transcriptional leakage, the shifting of protein distribution in model M5 was kept at minimum compared to the shift observed in M1 (Fig 2A and 2B). As a result, the X transcriptional inhibition of Z, for example, was also altered minimally in model M5 (Fig 2B,

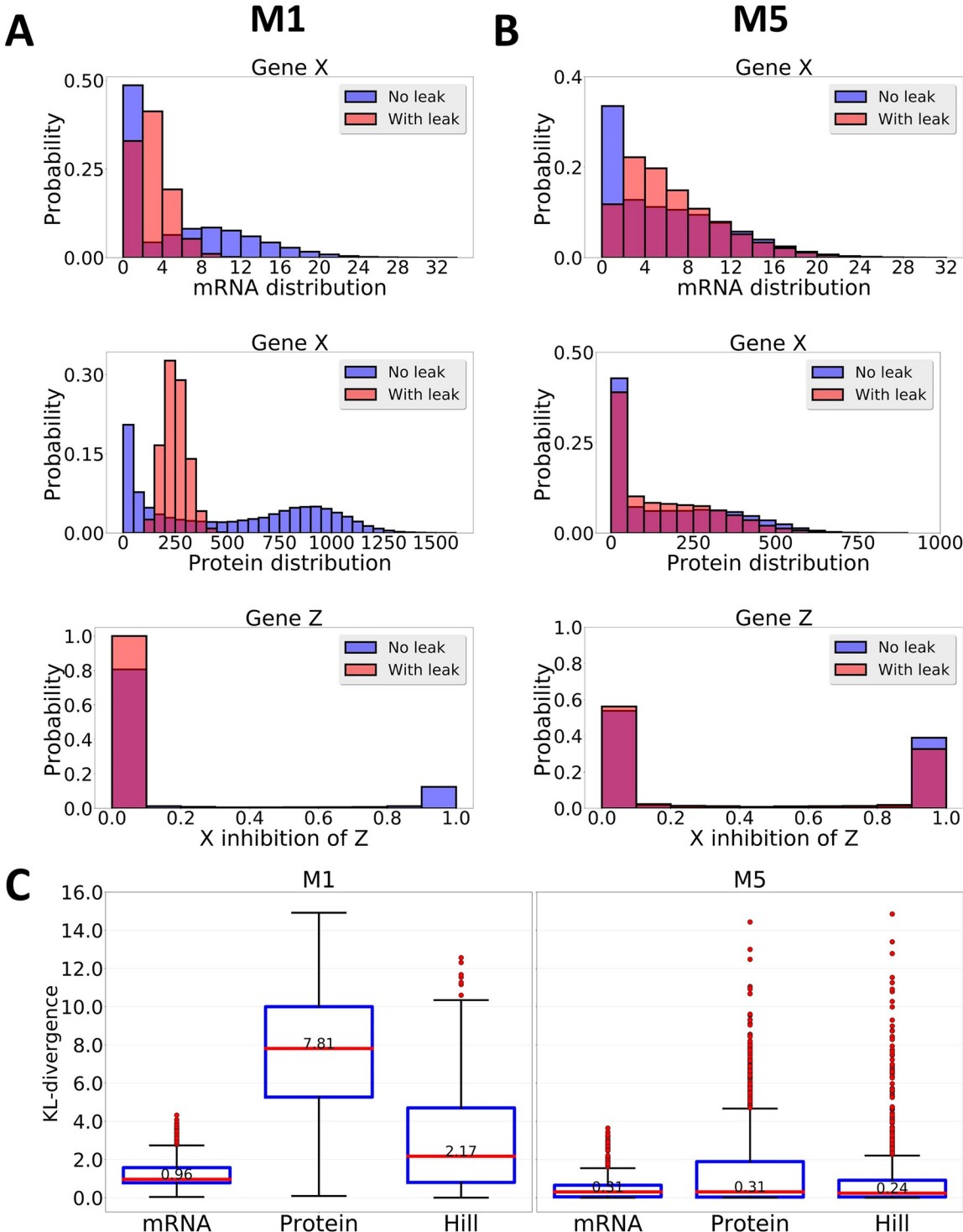

**Fig 2. Changes in the dynamics of M1 and M5 in the presence of basal leakage.** (A, B) The distribution of mRNA (upper panel), protein (middle panel), and transcription activity (the Hill function) of X inhibition of Z (lower panel) for one randomly chosen parameter set. (C) The Kullback–Leibler (KL) divergence between two distributions (with vs without basal) for all parameter sets in M1 (left) and M5 (right).

bottom). However, shifting the protein distribution of X greatly altered the X transcriptional inhibition of Z in M1 and locked it in the tight repressing state, which broke the oscillation (Fig 2A, bottom). Next, to test the generality of these observations, we calculated the Kullback–Leibler (KL) divergence between the two distributions (with and without basal) across different parameter sets [61]. Consistent with the observations above, we found that protein and hill function distributions were altered minimally despite of the observable shifting in the mRNA distribution (Fig 2C). Thus, our results suggested that combining both transcriptional and post-translational controls improve the robustness of the oscillator by controlling protein quantity when the rate of mRNA transcription is increased.

## Coupling degradation control to a repressilator using coherent or incoherent feed-forward loops gives better control to the leakage problem

To check the generality of our finding, we next expanded our model by taking into account other possible network structures and performed similar analyses as we did previously (Methods). For simplicity, but without loss of generality, we limited our analysis by keeping the repressilator as the core and targeted degradation (to be more specific 26S proteasome degradation) as the post-translation regulation. Other post-translational controls, such as phosphorylation, can also be modeled with similar mathematical forms to this degradation control, except for both the positive or negative effects to the system. With this formulation, we limited our analysis to four possible network structures from four different network motifs, which are type-3 coherent feed-forward loop (CFFL), type-2 incoherent feed-forward loop (IFFL), negative feedback (NF), and positive feedback (PF) (Fig 3A).

In the absence of transcriptional leakage, we found that the number of parameter sets that could produce sustained oscillation under the noisy condition were still lower in all tested model compared to M1 (Fig 3B). Moreover, the median value of normalized decay rates in CFFL, IFFL and NF were still higher than M1, which is consistent with our previous observation using M5 (Fig 3C and 3B). Of note, the PF model showed a better normalized decay rate compared to M1. This result is actually consistent with previous findings showing that coupling positive and negative feedback can create a more robust oscillation [62–65]. However, coupled positive and negative feedbacks must occur at transcription and post-translation levels. Otherwise, a decreased robustness in the oscillation was observed (M3 in Fig 1C).

In the presence of transcriptional leakage, we again found a similar observation as in model M5 for CFFL and IFFL models, such that 62.2% and 28% of parameter sets were still oscillating under 5% basal leakage, respectively. However, both PF and NF models failed to survive with transcriptional leakage (Fig 3C and S3 Fig). This finding is intriguing because both PF and NF models have similar degradation controls. To have better understanding of these results, we broke down the model into smaller network motifs and tested the effect of adding transcriptional leakage on both the input and output genes (Fig 4). Here, I and O denote the input and output genes, respectively, which are part of the core elements in the repressilator (either X, Y, or Z). The leakage was added in both I and O, while the degradation control (E) was kept the same (see Methods for detailed explanation). For simplicity, but without loss of generality, the production and the degradation rates for all genes were fixed at 1, with hill coefficient of 8. Since we are interested in analyzing the effect of adding basal leakage and degradation control on the downstream gene (O), we fixed any threshold value from I or O to E at 0.5, while scanning (from 0–1) for threshold value of I to O and E to O (S4–S7 Figs). However, for presentation purposes, we used 0.5 for both threshold value in Fig 4. The effect of post-translation control is considered advantageous if the turned-on basal expression in I leads to a similar status in O as if there were no basal added.

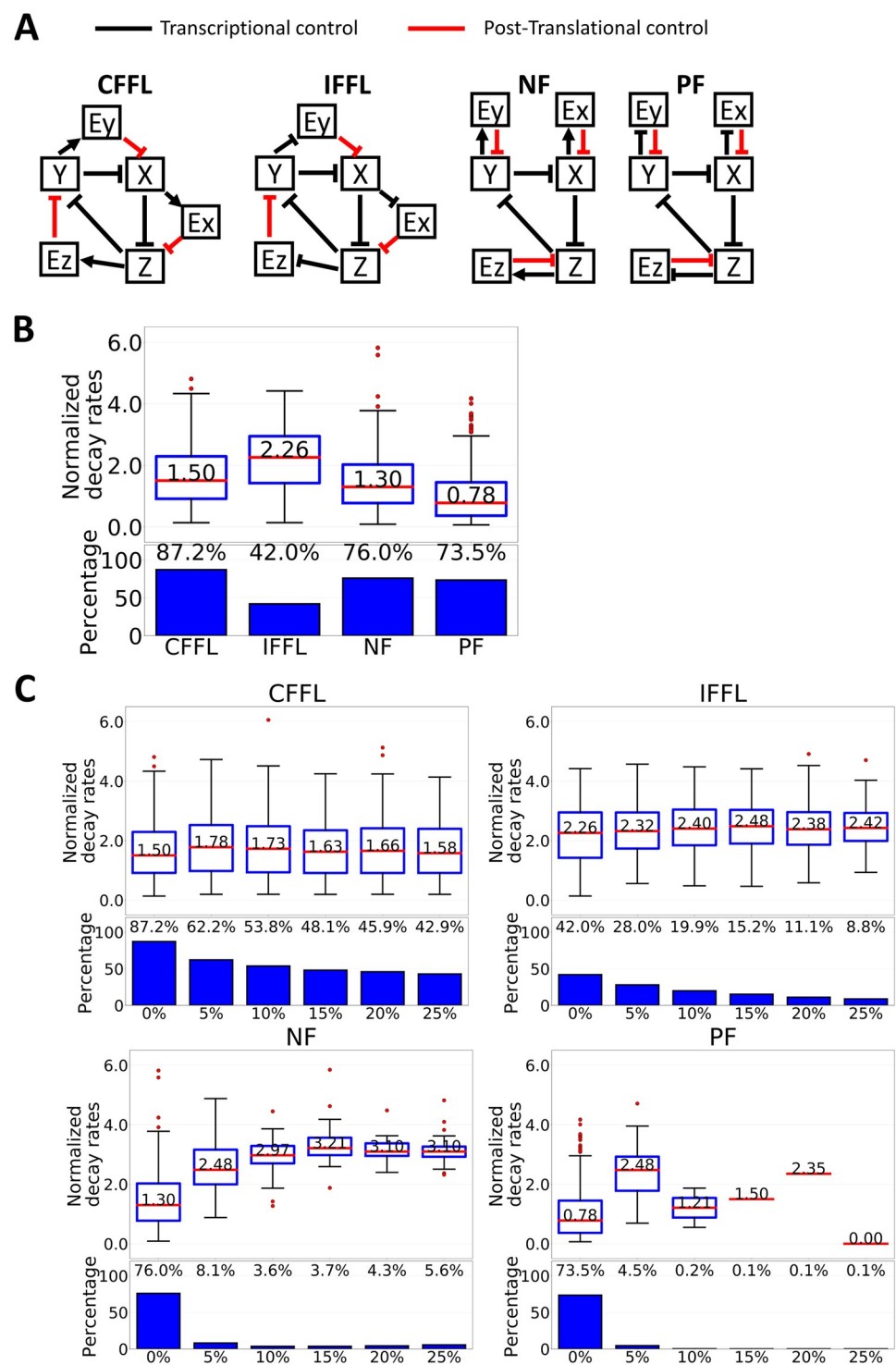

**Fig 3. Robustness test of several network motifs in coupling the degradation control to a repressilator.** (A) Schematic representation of the tested model. (B, C Upper panel) Box plot representing normalized decay rates of each tested model for 1000 randomly select parameter sets in the absence (B) or presence (C) of transcriptional leakage. Red lines indicate the median, and box edges indicate the 25th (Q1) and 75th (Q3) percentiles. Whiskers are plotted at 1.5* (Q3-Q1). (B, C Lower panel) Percentage of parameter sets that showed sustained oscillation under stochastic simulations.

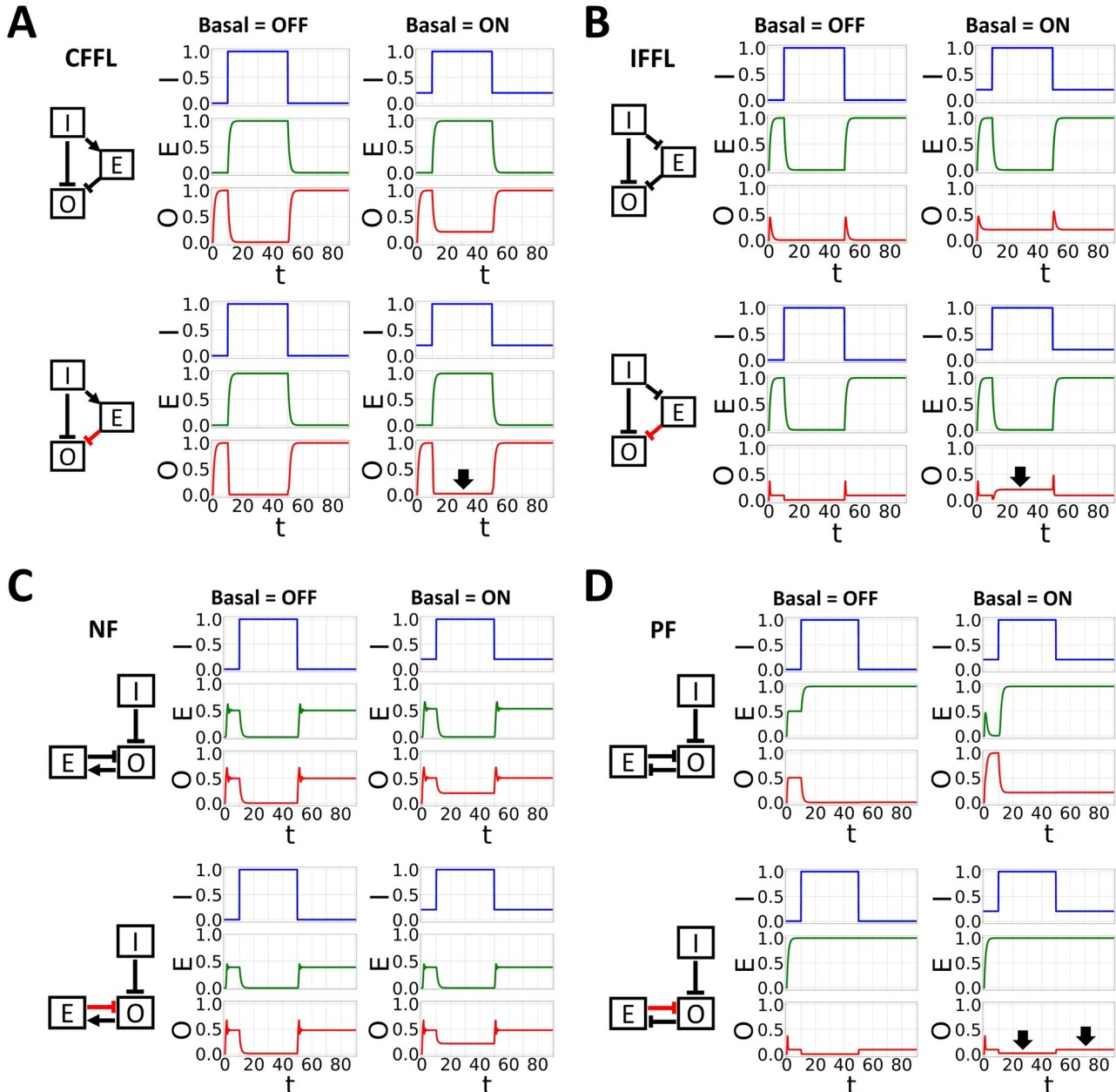

**Fig 4. The dynamics of (A) type-3 coherent feed-forward loop, (B) type-2 incoherent feed-forward loop, (C) positive feedback, and (D) negative feedback in controlling transcriptional leakage.** In upper panels, all genes are regulated through transcriptional controls. In lower panels, the E regulation of O was changed to a degradation control (indicates by red arrow). Black thick arrow highlights the filtering effect of a degradation control in the given network motif.

As seen in Fig 4, the CFFL motif is able to control the leakage when the input gene is ON, because E is only ON when I is ON. Hence, it can push back the transcriptional leakage that occurs in the output gene (Fig 4A). Furthermore, this observation is consistent across many different parameter combinations as shown in S4 Fig. In contrast with this observation, gene E in the IFFL motif is expressed only when gene I is OFF. Thus, IFFL can only control the leakage in O when I is OFF (Fig 4B and S5 Fig). For the PF motif, the leakage was actually controlled in both ON and OFF states of I (Fig 4C and S6 Fig). However, PF motif has another

problem, in which gene O is mostly kept in the OFF state (S6 Fig basal OFF panel). To have a higher amplitude of O, the PF motif requires the E regulation of O (K_Oe) to be weak (> 0.8, S6 Fig). However, when the E regulation of O is weak, the ability of the degradation control to push back the basal leakage is also weaker. Hence, we see no filtering ability in our PF model (Fig 3C). For the NF motif, we also cannot find any filtering ability of the degradation control on the transcriptional leakage (Fig 4D and S7 Fig). In this network, E only accumulated when the target gene is ON. Since the transcriptional leakage occurs mostly when O is OFF, the degradation control has almost no effect, which we can see in the NF model result above (Fig 3C). Thus, our results suggest that in the presence of transcriptional leakage, coupling 26S proteasome degradation using either coherent or incoherent feed-forward loops can help the repressilator to oscillate robustly.

## Our general models behave similarly to the increase variability observed in E3 ligase mutant in plant and mammalian clocks

From our observations above, we speculate that the 26S proteasome degradation could be important for the circadian clock to cope with transcriptional leakage and achieve robust oscillations in cells. To demonstrate this idea, we tried to compare the dosage-dependent effect of the *TOC1* and *PRR5* degradation control, *ZEITLUPE* (*ZTL*), on the dynamics of the *Arabidopsis* clock [30]. In 2004, Somers *et al.* showed that *ZTL* level controlled the amplitude and period of circadian oscillations [30]. Moreover, it also showed that a different *ZTL* dosage affected the robustness of the oscillation, which could be seen from a higher relative amplitude error (RAE) value (Fig 5A). We hypothesized that the increase in RAE is due to the shifting of the protein level in degradation control, where the system failed to push back the basal leakage under the mutant condition but greatly reduced the amplitude of *TOC1* and *PRR5* proteins under high continuous overexpression.

While it is impossible to directly verify our hypothesis without a proper model involving ZTL at the tissue level, since much of the regulation and mechanism are still unknown, we aimed to see if similar dosage-dependent effects can be obtained with our general model M5. Here, we varied the X post-translational regulation of Z by mutating or constantly overexpressing it during simulations (Methods). Under the mutant condition, the decay rates were generally increased (Fig 5C, upper panel) and the number of parameter sets that yielded oscillations were reduced from 28.1% to 5.4% under 10% leakage (S8A Fig). This result is similar to the experimental observation and was expected because the mutation of the X degradation control of Z will increase the basal expression of Z due to transcriptional leak. Hence, the system will suffer like the model M1 we described above (Fig 2).

However, the results of the overexpression condition are less straightforward than the mutant condition. We found that similar to the experimental result, the trend of decay rates was also increasing whereas the number of parameter sets that yielded oscillation also decreased along with the level of overexpression (Fig 5C and S8A Fig). Intuitively, one may expect that with more degradation control, we will see a more robust oscillation since the system can have better control on the transcriptional leakage. Hence, to have a better understanding of these results, we again tried to study the dynamics of the system by comparing the distribution of the overexpression mutant with the wild-type condition (Methods). Here, we found that adding a constant amount of targeted degradation will have a larger effect on the amplitude of protein Z rather than keeping the basal level of Z in the very low level. This decrease in Z protein level further affects the Z degradation control of Y, which increases the Y protein level. After that, the increase in Y protein greatly reduced the expression of gene X (S9 Fig), which prevented the system from having robust oscillation. Furthermore, to rule out a

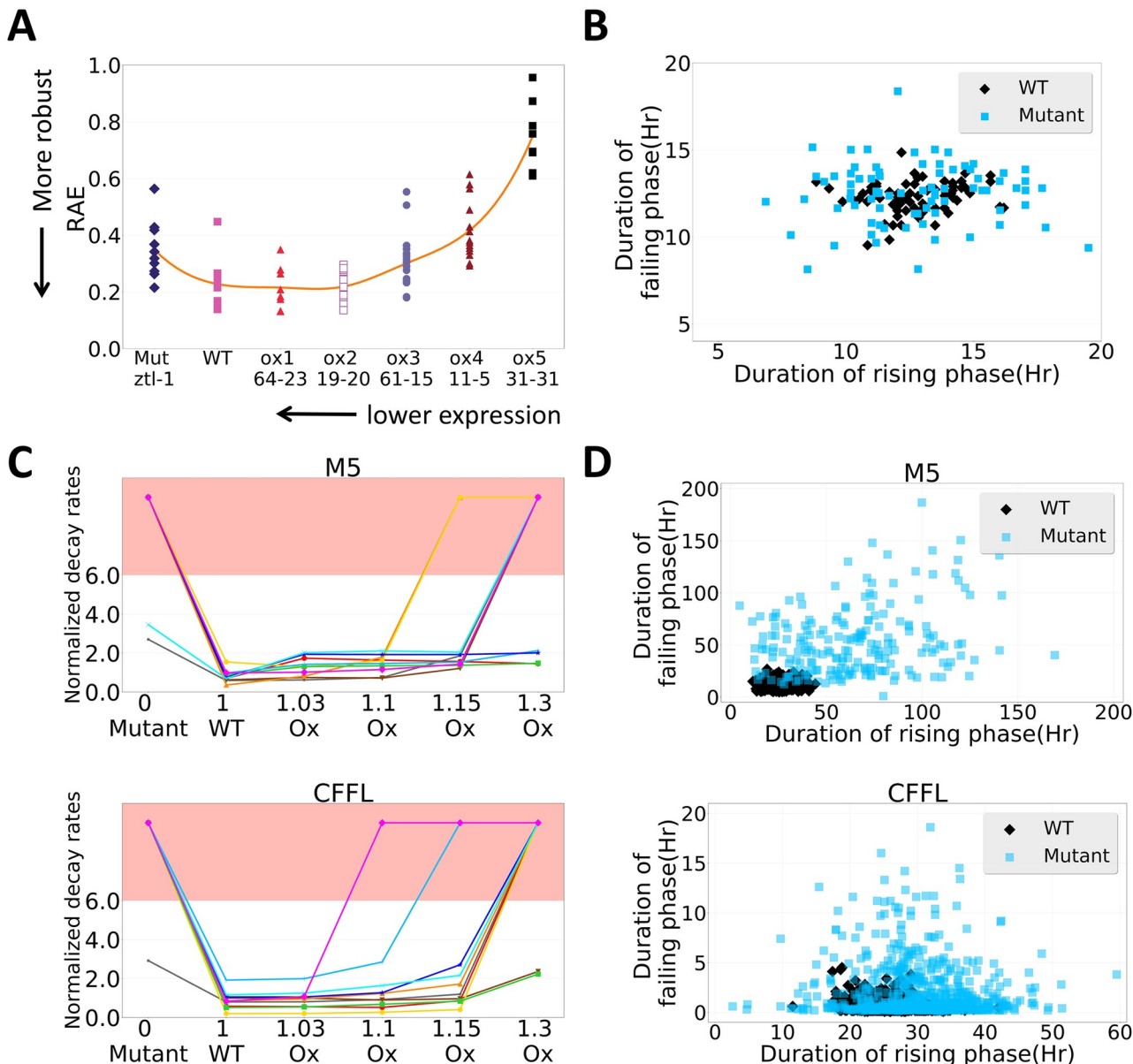

**Fig 5. Our general models were able to replicate several observed experimental results.** Experimental result of dosage-dependent effect of proteasome degradation control (*ZTL*) to the robustness of the oscillator. Data were obtained and redrawn from Somers *et al.* 2004 [30]. (B) Experimental result of *β-Trcp2* mutant cells showed unstable rhythms and reduced amplitude of *PER* gene oscillations. Data were obtained and redrawn from D'Alessandro *et al.* 2017 [47]. (C) The effect of changing the degradation control in model M5 (upper panel) and CFFL (lower model) during our simulations. Each line represents one parameter set. The red part of the plot indicates the non-oscillating region. (D) The effect of mutating the degradation control to the rising and failing phases of model M5 (upper panel) and CFFL (lower panel) during our simulations. The plot was drawn from one randomly selected parameter set.

possible limitation of using a simple model (such as M5), we performed a similar analysis using a more elaborate model, CFFL. Here, we again obtained a similar result such that the robustness of the oscillation was reduced when we mutated or overexpressed the Ex degradation control of Z (Fig 5C, lower panel, and S8B Fig).

Lastly, we also tried to link insights derived from our model study to the effect of mutating the E3 ligase gene on the dynamics of mammalian clock. In 2017, D'Alessandro *et al.* showed

that mutation of the E3 ligase gene, *β-Trcp2*, will perturb the balance of *PERIOD* (*PER*) degradation, which makes the clock unstable (Fig 5B) [47]. As we discussed above, we also observed unstable oscillation when we changed the degradation control in our models. We believe that our previous insight can be used to explain the observed experimental results in the mammalian clock. Hence, we again performed a similar analysis as we did previously and found that mutation of the degradation control will indeed alter the duration of the rising and failing phase of the oscillation (Fig 5B and 5D). These results indicate that the expression leakage and protein degradation could influence the mammalian clock in a similar way.

## Discussion

### Coupling E3 ligase to a negative feedback loop using feed-forward loop is commonly seen in the *Arabidopsis* circadian clock

Our results showed that combining two types of regulation using feed-forward loop can create a robust oscillator. In the *Arabidopsis* clock, several E3 ligases and their respective targets have been successfully identified. The first identified E3 ligase was an F-box protein, *ZEITUPLE (ZTL)*. *ZTL* was reported to be involved in target degradation of both *PRR5* and *TOC1* proteins [29,38]. Although *ZTL* mRNA is consecutively expressed, its protein still oscillated with three-fold change in amplitude [66]. This oscillation may be mediated through a protein–protein interaction of *ZTL* with *GIGANTEA (GI)* protein [67]. Also, *CCA1* and *LHY* can bind and repress the expression of *GI*. Hence, together with ZTL, they form a feed-forward network motif (Fig 6A).

Next, a RING-type E3 ligase protein, *constitutive photomorphogenesis 1 (COP1)*, has also been reported to degrade other important clock genes, *ELF3* and *ELF4* [40,70]. *ELF3*, *ELF4* and *LUX* have been reported to form a protein complex, called *evening complex (EC)*, and the loss-of-function mutant of any of these three proteins will lead to arrhythmic behavior [26,71–73]. Previously, *CCA1* and *LHY* were reported to bind and repress the expression of both *ELF4* and *ELF3* [15,74,75]. However, little is known about the regulator of *COP1* proteins. From our results, we hypothesized that the clock system could have the advantages that we mentioned above if *CCA1* and *LHY* can directly (or indirectly) regulate COP1. Hence, we performed a quick analysis combining ChIP-seq data for *CCA1* [15,74] and *LHY* [75] with TF binding prediction tools, PlantTFDB [68] and PlantPAN3 [69], to predict the direct or indirect regulation of *CCA1/LHY* to *COP1*. Interestingly, our analysis suggests that a well-studied transcriptional factor, *TCP21 (TCP21/CHE)*, could bind to the *COP1* promoter region and both *CCA1* and

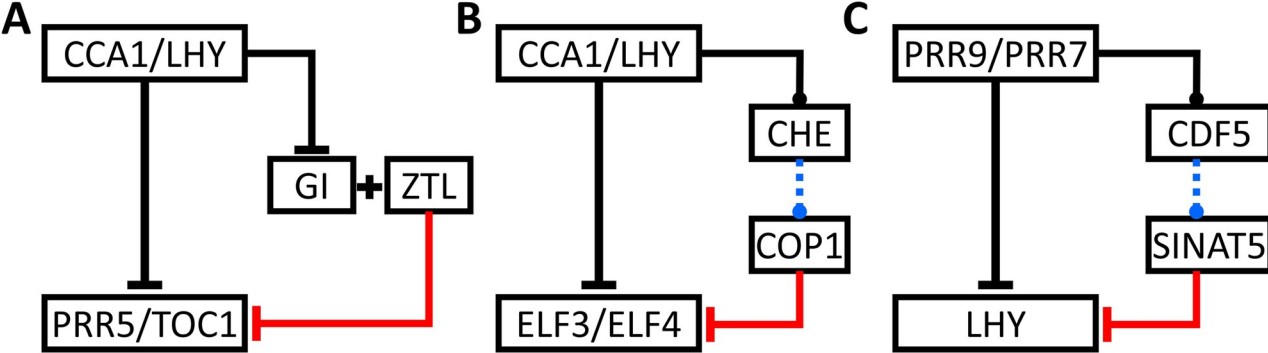

**Fig 6. Currently known (A) or predicted (B and C) regulators of E3 ligase in the plant's clock form a feed-forward network motif.** Black and blue lines represent transcriptional regulations, and the red line represents the degradation control. Solid lines indicate that the data were derived from experimental results, whereas dashed lines indicate that the data were derived from TF binding prediction tools, PlantTFDB [68] and PlantPAN3 [69].

*LHY* could also bind to the *TCP21* promoter region. Together, they form an indirect feed-forward network motif (Fig 6B).

Finally, another RING-type E3 ligase protein, *SINAT5* was also reported to interact and degrade *LHY* protein in the plant [35]. However, similar to *COP1*, little is known regarding the regulator of *SINAT5*. Hence, we performed a similar analysis as we did for *COP1*, only now we used ChIP-seq data for *PRR9* and *PRR7* [76]. We found that *cyclic dof factor5 (CDF5)* could bind to the SINAT5 promoter region, whereas both *PRR9* and *PRR7* were found in the *CDF5* promoter region. Together, they form another indirect feed-forward network motif (Fig 6C).

Although these predictions are yet to be verified experimentally, they provide other evidence that strengthens our simulation results. Previously, several mathematical models have shown that the core oscillator of the *Arabidopsis* clock consists of four groups of genes, the early morning phase genes (*CCA1/LHY*), the daytime/noon phase genes (*PRR9/PRR7*), the afternoon/dusk phase genes (*PRR5/TOC1*), and the nighttime/midnight phase genes (*EC*) [22–24,48]. Among these four groups, three may be regulated by coupled repression and degradation through feed-forward loops (Fig 6). To the best of our knowledge, the corresponding E3 ligases for the noon phase genes, *PRR9* and *PRR7*, are still elusive. However, several studies have shown that both genes are still regulated by the 26s proteasome degradation pathway [37,77]. It will be interesting to see whether these two genes are also under similar regulation. Thus, these findings imply that coupling E3 ligase to a negative feedback using a feed-forward loop is probably common in the *Arabidopsis* clock.

## Transcriptional leakage is commonly seen in cells

Transcription leakage (or basal transcription) commonly occurs in cells. For example, several systems like CpxR in *Escherichia coli* and VirG in *Agrobacterium tumefaciens* have been reported to have high basal expression [78–80]. Moreover, Yanai *et al.* also showed that the expression of many tissue-specific genes can "overflow" into neighboring genes that have no function in the respective tissue [59]. This observation implies that leakiness of the promoters may be evolutionarily neutral [60].

In this study, we showed that leakage can be a serious issue in oscillating systems like the circadian clock. In cells, this problem is probably handled by having a dual transcription and post-translation control, which gives another layer of regulation for cells to deal with the leakage problem. Actually, in line with our findings, many synthetic systems have emphasized the importance of combining both transcription and post-transcription/translation controls in creating a robust oscillator [79,81–84]. For instance, Tigges *et al.* showed that a synthetic mammalian oscillator based on transcription (sense) and a post-transcription control (antisense) can create an autonomous, self-sustained and tunable oscillator [82]. In another study, Danino *et al.* showed that coupling transcription (LuxR) and degradation control (AiiA) with global intercellular coupling (AHL) generates synchronized, robust oscillation [83].

Furthermore, several theoretically studies have also highlighted the importance of degradation control in oscillation system. For instance, in 2006, Krishna, *et al.* showed that depending on the saturation of active degradation rate of IκB (an inhibitor of NF-κB), the oscillation of NF-κB can be extremely robust to variation of parameters [85]. In a 2007 work, Wong *et al.* also showed that adding non-linear degradation term significantly enlarges the parameter space for oscillation and thus enhances the robustness of synthetic gene metabolic oscillator [86]. Recently, Clamons and Murray showed that their CRISPRi repressilator model is very sensitive to basal leak, such as adding 1% leak will destroy the oscillation. They have also found that this loss of oscillations by leak can be offset by adding active degradation to dCas [87].

Hence, our findinga are in line with these observations, showing that coupling transcriptional and degradation control may be important in controlling transcriptional leakage in cells.

Although leaky expression is often considered as unwanted noise in cells, it can also be useful in some systems. For instance, in the competence development of *Bacillus subtilis*, basal expression of the master regulator (*comK*) must pass a critical threshold in order to trigger its auto-activation, which will lead to a bistable pattern in *B.subtilis* cells. This bistability will further create heterogeneity in cell populations, which can benefit the population by providing better-adapted phenotypic variants for a given perturbation [88]. In contrast with this finding, Ingolia and Murray in 2007 showed a different role of basal expression in creating bistability. Using a budding-yeast pheromone response system, they could make this system bistable by reducing the basal expression [89]. Hence, depending on the evolutionary process, basal expression can be considered an important aspect or a noise in different biological systems.

## Limitation and further direction

Stochastic simulation algorithm (SSA, [54]) is still the most common algorithm for studying the dynamics of noisy biological system. The main advantages of using this algorithm is that it gives an exact simulation of the stochastic process described by the chemical master equation (CME). However, calculating every reaction event is costly, which make this algorithm not particularly suited for studying large network. To circumventing this problem, quasi steady-state approximation (QSSA), such as Michaelis-Menten or the Hill function, is frequently used to reduce the complexity of the networks. It has been reported previously that stochastic QSSA results give good agreement with the full networks for enzyme reactions and circadian oscillators [90–93]. Furthermore, stochastic QSSA is also the most widely used algorithm due to its simplicity and general applicability [94–97]. Thus, we adopted this simplification in our model to significantly speed up our simulation process, such as employing the Hill function for transcription and translational control, in mimicking the saturation effect in the up-stream regulator.

However, like other simplification process, QSSA also has limitation. Several studies recently have shown examples where using approximation QSSA can lead to considerable errors in the stochastic simulation [98–101]. Since then, a lot of effort has been done to find rigorous condition to check the validity of stochastic QSSA [99,100] or deriving an exact QSS of fast species rather than using approximate QSS [101]. Hence, depending the purpose of the study, one need to choose more carefully which simplification technique they should use.

Furthermore, it has been reported previously that in some system, stochasticity can prevent the damping of oscillations or even create sustained oscillations. Stochastic resonance is one class of such phenomena where systems oscillate due to noise [102–104]. In biological systems, stochastic resonance was reported mainly in excitable systems like in neurons or the P53-Mdm2 oscillation which is in respond to external stimulus and the oscillation does not need to be robust and stable [105–107]. However, in this study, those parameter sets, which only oscillate under noise, have been ignored. In general, we believe that circadian clock's oscillations should be robust and it should not depend on noise. Hence, picking parameter sets that oscillate during ODE simulations is a reasonable strategy. However, it remains unknown whether stochastic resonance or similar phenomena could play a role in circadian clocks.

Next, there are also many ways to model gene regulations in the cells. Currently, Hill function is the most common way to describe regulation process many plant circadian clock models [22–24,48,108,109]. However, a new class of models with protein sequestration-base repression has been introduced recently to model the circadian process in mammalian cells [47,110]. It was shown previously that this new class of models differs from Hill-based

repression (as discussed thoroughly in [111]). Therefore, depending on the key regulation process, one may choose which method is suited best for their system of interest.

Previously, it has been shown that IFFL can help turning *PRR9/7* into a rapid switcher, which is important for replicating the correct dynamics of the clock system [23]. In this study, we showed that coupling 26S proteasome degradation using feed-forward loops (either coherent or incoherent) can help the repressilator to oscillate robustly in the presence of transcriptional leakage. Although both results seem similar, it has some fundamental differences. For transcriptional IFFL, it is important for the system to have a pulse-like expression, which may help it to response rapidly in facing sudden changes of environmental signal, while for translational control such requirement is not needed. In fact, our results showed that CFFL, which do not generate pulse-like expression, has better performance than IFFL in coupling the degradation control. Hence, it will be interesting to combine these two models in the future and test whether combining these properties can lead to new, more interesting, features, which can further enhance our understanding of the *Arabidopsis* circadian clocks.

## Methods

### Model representation

All models used in this study are described by a set of ordinary differential equations (ODEs) for the simulation under continuous light. In general, each gene was represented as:

$$\frac{dP}{dt} = \beta.Hill - \gamma P \tag{1}$$

where P represents the dimensionless concentration, which can be any genes depending on the model. Here, $\beta$ represents the total production rate, whereas $\gamma$ is the total degradation rate. The nondimensionalization process involved choosing a constant value for each component, denoted as $P_0$. $P_0$ was defined as the maximum steady state of gene P, which is the ratio of total production rates over total degradation rates. Hill represents the Hill function, which describes the effects of upstream regulation as

$$Hill_{act} = \frac{[Activator]^n}{\kappa^n + [Activator]^n} \tag{2}$$

for the activating process and

$$Hill_{rep} = \frac{\kappa^n}{\kappa^n + [Repressor]^n} \tag{3}$$

for the repression process. As mentioned previously, the Hill input function is a monotonic, S-shaped function, which is used to describe the effect of the transcription factor on the transcription rate of its target gene [112]. Here, $\kappa$ represents the concentration of the activator or repressor needed to achieve half maximal effect. It is related to the binding affinity between the transcriptional factor gene and its site on the promoter region [112]. The *n* represents the Hill coefficient that governs the steepness of the input function, which is related to sensitivity of the process in the cell. Employing *n* allows us to describe ultra-sensitivity of many cellular processes, such as multisite phosphorylation, stoichiometric inhibitor, cooperativity, reciprocal regulation, and substrate competition [113]. Previously, several clock proteins have been shown to form a dimer [27,114,115]. Because of this reason, many mathematical models set *n* = 2, which corresponds to this dimerization process [22,48,116]. However, in this study, we allowed *n* to be > 2 to accommodate other possible regulations (Table 1). Following previous

**Table 1. Search ranges for parameters.**

| Parameters | Range | Units | Search scale |
|---|---|---|---|
| $\gamma_{basal}$ [1] | 0.01–10 | 1/Hr | Logarithm |
| $\gamma_{deg}$ | 1–1000 | 1/Hr | Logarithm |
| $\kappa$'s (in all Hill functions) | 1–1000 | Dimensionless | Logarithm |
| $n$'s (in all Hill functions) [2] | 2–8 | Dimensionless | Linear |

[1] $\beta$'s are set to be 1000 times higher than the corresponding $\gamma$'s for a dimensionless unit for the concentration of each gene and for a reduction in number of parameters.
[2] Only integer value was used for this hill coefficient.

studies, we used an 'AND' gate to describe a combination of two or more source of regulations, where the two Hill functions are multiplied [22,24,48].

Finally, we also fixed the maximum possible steady-state concentration of each component to 1000 molecules per cell by assuming the Hill function as its maximum possible value, 1. In this way, the maximum production rates ($\beta$'s) were fixed to be 1000 times higher than the total degradation rates ($\gamma$'s). The time $t$ in the current work was in the unit of hours. After obtaining a regularly oscillating parameter set in the wild-type setting, we re-scaled all the time-related parameters such that the oscillation period is 24 hr.

**Model M1: Transcriptional control based repressilator.**

$$\frac{dX}{dt} = \beta_x.Hill_{rep\_Y} - \gamma_x.X, \tag{4}$$

$$\frac{dY}{dt} = \beta_y.Hill_{rep\_Z} - \gamma_y.Y, \tag{5}$$

$$\frac{dZ}{dt} = \beta_z.Hill_{rep\_X} - \gamma_z.Z. \tag{6}$$

**Model M2: Post-translational control based repressilator.**

$$\frac{dX}{dt} = \beta_x - (\gamma_{basal\_X} + \gamma_{deg\_X}.Hill_{act\_Y}).X, \tag{7}$$

$$\frac{dY}{dt} = \beta_y - (\gamma_{basal\_Y} + \gamma_{deg\_Y}.Hill_{act\_Z}).Y, \tag{8}$$

$$\frac{dZ}{dt} = \beta_z - (\gamma_{basal\_Z} + \gamma_{deg\_Z}.Hill_{act\_X}).Z. \tag{9}$$

We assumed that the target degradation happened much faster than basal degradation. Hence, the searching space for $\gamma_{deg}$ was set 2 orders higher (Table 1).

**Model M3: Transcriptional control based repressilator with additional positive feedback loop.** In model M3, Eq (4) was modified into Eq (10),

$$\frac{dX}{dt} = \beta_x.Hill_{rep\_Y} + \beta_{pos}.Hill_{act\_X} - \gamma_x.X, \tag{10}$$

whereas the other two equations remained the same (Eqs (5) and (6)). There were two additional parameters (the corresponding production rates ($\beta_{pos}$) and $\kappa$ value in the new Hill function $Hill_{act\_X}$) added in this model. The parameters were treated similarly as those in M1.

**Model M4: Post-translational control based repressilator with additional positive feedback loop.**   For model M4, Eq (7) was modified into Eq (11),

$$\frac{dX}{dt} = \beta_x + \beta_{pos}.Hill_{act\_X} - (\gamma_{basal\_X} + \gamma_{deg\_X}.Hill_{act\_Y}).X, \tag{11}$$

while other two equations (Eqs (8) and (9)) were kept the same. There were again two additional parameters added in this model. The parameters were treated similarly as those in M2.

**Model M5: Coupled transcriptional and post-translational control-based oscillator.**
For model M5, we combined the regulation of model M2 into model M1 and described it as:

$$\frac{dX}{dt} = \beta_x.Hill_{rep\_Y} - (\gamma_{basal\_X} + \gamma_{deg\_X}.Hill_{act\_Y}).X, \tag{12}$$

$$\frac{dY}{dt} = \beta_y.Hill_{rep\_Z} - (\gamma_{basal\_Y} + \gamma_{deg\_Y}.Hill_{act\_Z}).Y, \tag{13}$$

$$\frac{dZ}{dt} = \beta_z.Hill_{rep\_X} - (\gamma_{basal\_Z} + \gamma_{deg\_Z}.Hill_{act\_X}).Z. \tag{14}$$

Although the transcriptional and post-translational regulations are usually controlled by different genes, for simplicity, we assumed that it might be originated from the same transcriptional regulator. For instance, to post-translationally modify protein Z, gene X might actually need to activate another protein, X', that can bind to and modify protein Z. However, if we assume that the expression of gene X' is solely dependent on gene X and this activation process happens fast enough, we can omit gene X' in our model and replace this regulation by a hill function. By doing so, we can greatly reduce the complexity of the model M5.

**Various variant of coupled transcription and post-translation model.**   To overcome the limitations of using simplified models, we next tried to elaborate our model by taking into consideration several types of regulations that have been commonly discussed in the other studies. Here, we tested four different network motifs, Coherent feed-forward loop (CFFL), Incoherent feed-forward loop (IFFL), negative feedback loop (NF), and positive feedback loop (PF).

1. *Coherent Feed-Forward Loop (CFFL)*

$$\frac{dX}{dt} = \beta_x.Hill_{rep\_Y} - (\gamma_{basal\_X} + \gamma_{deg\_X}.Hill_{act\_Ey}).X, \tag{15}$$

$$\frac{dY}{dt} = \beta_y.Hill_{rep\_Z} - (\gamma_{basal\_Y} + \gamma_{deg\_Y}.Hill_{act\_Ez}).Y, \tag{16}$$

$$\frac{dZ}{dt} = \beta_z.Hill_{rep\_X} - (\gamma_{basal\_Z} + \gamma_{deg\_Z}.Hill_{act\_Ex}).Z. \tag{17}$$

$$\frac{dE_x}{dt} = \beta_{Ex}.Hill_{act\_X} - \gamma_{Ex}.E_x, \tag{18}$$

$$\frac{dE_y}{dt} = \beta_{Ey}.Hill_{act\_Y} - \gamma_{Ey}.E_y, \tag{19}$$

$$\frac{dE_z}{dt} = \beta_{Ez}.Hill_{act\_Z} - \gamma_{Ez}.E_z, \tag{20}$$

2. *Incoherent Feed-Forward Loop (IFFL)*

$$\frac{dX}{dt} = \beta_x.Hill_{rep\_Y} - (\gamma_{basal\_X} + \gamma_{deg\_X}.Hill_{act\_Ey}).X, \tag{21}$$

$$\frac{dY}{dt} = \beta_y.Hill_{rep\_Z} - (\gamma_{basal\_Y} + \gamma_{deg\_Y}.Hill_{act\_Ez}).Y, \tag{22}$$

$$\frac{dZ}{dt} = \beta_z.Hill_{rep\_X} - (\gamma_{basal\_Z} + \gamma_{deg\_Z}.Hill_{act\_Ex}).Z. \tag{23}$$

$$\frac{dE_x}{dt} = \beta_{Ex}.Hill_{rep\_X} - \gamma_{Ex}.E_x, \tag{24}$$

$$\frac{dE_y}{dt} = \beta_{Ey}.Hill_{rep\_Y} - \gamma_{Ey}.E_y, \tag{25}$$

$$\frac{dE_z}{dt} = \beta_{Ez}.Hill_{rep\_Z} - \gamma_{Ez}.E_z. \tag{26}$$

3. *Negative Feedback (NF)*

$$\frac{dX}{dt} = \beta_x.Hill_{rep\_Y} - (\gamma_{basal\_X} + \gamma_{deg\_X}.Hill_{act\_Ex}).X, \tag{27}$$

$$\frac{dY}{dt} = \beta_y.Hill_{rep\_Z} - (\gamma_{basal\_Y} + \gamma_{deg\_Y}.Hill_{act\_Ey}).Y, \tag{28}$$

$$\frac{dZ}{dt} = \beta_z.Hill_{rep\_X} - (\gamma_{basal\_Z} + \gamma_{deg\_Z}.Hill_{act\_Ez}).Z. \tag{29}$$

$$\frac{dE_x}{dt} = \beta_{Ex}.Hill_{act\_X} - \gamma_{Ex}.E_x, \tag{30}$$

$$\frac{dE_y}{dt} = \beta_{Ey}.Hill_{act\_Y} - \gamma_{Ey}.E_y, \tag{31}$$

$$\frac{dE_z}{dt} = \beta_{Ez}.Hill_{act\_Z} - \gamma_{Ez}.E_z. \tag{32}$$

4. *Positive Feedback (PF)*

$$\frac{dX}{dt} = \beta_x.Hill_{rep\_Y} - (\gamma_{basal\_X} + \gamma_{deg\_X}.Hill_{act\_Ex}).X, \tag{33}$$

$$\frac{dY}{dt} = \beta_y.Hill_{rep\_Z} - (\gamma_{basal\_Y} + \gamma_{deg\_Y}.Hill_{act\_Ey}).Y, \tag{34}$$

$$\frac{dZ}{dt} = \beta_z.Hill_{rep\_X} - (\gamma_{basal_Z} + \gamma_{deg\_Z}.Hill_{act_{Ez}}).Z, \tag{35}$$

$$\frac{dE_x}{dt} = \beta_{Ex}.Hill_{rep\_X} - \gamma_{Ex}.E_x, \tag{36}$$

$$\frac{dE_y}{dt} = \beta_{Ey}.Hill_{rep\_Y} - \gamma_{Ey}.E_y, \tag{37}$$

$$\frac{dE_z}{dt} = \beta_{Ez}.Hill_{rep\_Z} - \gamma_{Ez}.E_z. \tag{38}$$

## Searching, propagation, and selection process

All independent parameters in each model were obtained by random searches, propagated, and screened for regular oscillation. Here, random parameters were drawn from a uniform distribution in the log scale because we do not have any prior knowledge of the real parameter distribution. Hence, we assume all random numbers are equally plausible to be picked. The search was performed at a logarithmic scale across three orders of magnitude, for $\gamma$'s and $\kappa$'s, and a linear scale for $n$ (Table 1). Here, a parameter set is selected if the trajectory can oscillate regularly, defined by examining the period and amplitude change in each cycle. We calculated the relative difference in period and amplitude change for each cycle, defined as $\frac{|x_1 - x_2|}{min(x_1, x_2)}$, where $x_1$ and $x_2$ are the period or amplitude calculated from two consecutive cycles. An acceptable regular oscillation was defined as that with less than 5% relative change for more than 10 cycles. For all searching, we used similar initial conditions, which is 10% of maximum possible steady-state concentration. Finally, all simulations were performed by using Python 3.6 (Anaconda 4.4.0).

## Stochastic simulations

For all models, the stochastic simulation was done using Gillespie algorithm [53,117]. Here, each gene is described in three different levels: gene, mRNA, and protein levels. For instance, for any gene A in model M5 (can be either X, Y, or Z), the ODE equation can be transformed into:

$$\frac{dG_A}{dt} = k_{g\_A}.(1 - G_A) - \gamma_{g\_A}.G_A, \tag{39}$$

$$\frac{dM_A}{dt} = k_{m\_A}.G_A.Hill - \gamma_{m\_A}.M_A, \tag{40}$$

$$\frac{dP_A}{dt} = k_{p\_A}.M_A - \left(\gamma_{pA} + \gamma_{deg\_A}.Hill\right).P_A, \tag{41}$$

where $G_A$ is the fraction of active gene A for transcription, while $M_A$ and $P_A$ is the amount of mRNA and proteins of gene A, respectively. $k$ represents the activation or production rates and $\gamma$ is the deactivation or degradation rate. It has been shown previously that both mRNAs and protein are produced in bursts [118,119]. Following previous study, we defined the ratio between $\frac{k_m}{\gamma_g}$ as the mean burst frequency ($B_m$) and the ratio of $\frac{k_p}{\gamma_m}$ as mean burst size ($B_p$) [120]. For simplicity, but without loss of generality, we fixed our $B_m$ and $B_p$ into 2 and 10, respectively. Last, we also set $\gamma_g$ and $\gamma_m$ at 100 and 10 times of $\gamma_p$, , which satisfies the condition of

**Table 2. Summary of parameter conversion (using gene X of model M5 as an example).**

| Propensity function | Corresponding values |
|---|---|
| $\gamma_{p\_X}$ | $\gamma_x$ [1] |
| $\gamma_{m\_X}$ | $\gamma_{p\_X}$ * 10 |
| $\gamma_{g\_X}$ | $\gamma_{p\_X}$ * 100 |
| $k_{g\_X}$ | $\frac{\beta_x^{1}}{Bm*Bp}$ |
| $k_{m\_X}$ | $Bm$ * $(k_{g\_X} + \gamma_{g\_X})$ |
| $k_{p\_X}$ | $Bp$ * $\gamma_{m\_X}$ |
| $Hill_{rep\_Y}$ | $Hill_{rep\_Y}$ [1] |
| $\gamma_{deg\_X}$ | $\gamma_{deg\_X}$ [1] |
| $Hill_{act\_Y}$ | $Hill_{act\_Y}$ [1] |

[1] Parameter value obtained from ODEs simulation.

burst production in both mRNA and protein levels as described previously [120]. By doing so, we can obtain all necessary propensity functions from our ODE equation without any additional parameter value to be searched (Table 2).

## Measuring the oscillations under stochastic simulations

There are several ways to measure the "goodness" of oscillatory behavior [121,122]. However, in this study, we make use of the autocorrelation function to measure the "goodness" of the oscillations. Let $P_{(M.\Delta t)}$ be a time series data, corresponding to one protein component in our simulations, with length of M times $\Delta t$ (in our simulation, $\Delta t$ is fixed into 0.1 Hr). Mathematically, the autocorrelation function of $P_{(M.\Delta t)}$ is defined as:

$$ACF(n\Delta t) = \frac{1}{(M-n)} \cdot \sum_{i=0}^{M-n} \frac{(P_i - \mu).(P_{i+n} - \mu)}{\sigma^2}, \tag{42}$$

where $n$ can be any value between 0 to M-1. In our simulation, we fixed our $n$ such that it is between 0 and 2400 (equivalent to 10 days). Using this equation, the maximum value of Eq (42) is when $ACF(0)$, which is 1. If $P_{(M.\Delta t)}$ is the output of deterministic simulation with sustained oscillation, then Eq (42) will also oscillate in a sustained manner. This kind of oscillation can be estimated by using Eq (43),

$$A + \cos\left(\frac{2\pi X}{T}\right), \tag{43}$$

where $X$ is the autocorrelation function value, and $T$ is the period of the oscillation. However, if $P_{(M.\Delta t)}$ describes a realization of a stochastic oscillatory process, Eq (42) will show a damped oscillation. In this case, this dampened oscillation can be estimated by Eq (44),

$$A + \cos\left(\frac{2\pi X}{T}\right)e^{-bX}, \tag{44}$$

where $b$ represents the damping rate or characteristic time of the decay of the autocorrelation function [122,123]. Notice that parameter b will still carry the time dimension. Hence, it can be nondimensionalized by dividing it with the period parameter, T. In this study, we can even further simplify this estimation process by just fitting the dampened autocorrelation function

with an exponential function,

$$normalized\ decay\ rate = A + e^{-bX'}, \tag{45}$$

where $X'$ represents all the peak value obtained in the autocorrelation function.

## Modified stochastic simulations

**Adding transcriptional leakage.** To add a transcriptional leakage in our stochastic simulations, we modified Eq (40),

$$\frac{dM_x}{dt} = k_{m_X}.G_x.\Big(basal(1 - Hill_{rep\_X}) + Hill_{rep\_X}\Big) - \gamma_{m\_X}.M_x, \tag{46}$$

and applied it to all genes in each model. Other equations (Eqs (39) and (41)) were kept the same. We argue that this representation is more realistic than simply adding a constant basal expression in the mRNA level, because with this modification, the maximum value of the hill function is kept.

**Mutation and overexpression of degradation control in M5.** For the overexpression test of model M5, we modified the X degradation control of gene Z ($Hill_{act\_X}$ in Eq (14)) from Eq (2) into Eq (47),

$$hill_{act\_X} = \frac{(X + constant\_OX)^n}{\kappa^n + (X + constant\_OX)^n} \tag{47}$$

with other equations kept the same. In this analysis, we cannot simply overexpress gene X, because it would also affect the gene X transcriptional regulation of gene Z ($hill_{rep\_X}$ in Eq (14)). Next, we varied the value of *constant_OX*, from 3% to 30%, based on the maximum possible steady state of ODE oscillation (which is 1000). For the null-mutant test, we simply set $\gamma_{deg\_Z}$ in Eq (14) to 0.

**Mutation and overexpression of degradation control in other models.** For the other models (CFFL, IFFL, NF, PF), the genetic perturbation tests were done by scaling up (for overexpression) or scaling down (for null-mutant) the production rates ($\beta$) of gene Ex (for CFFL and IFFL) or gene Ez (for NF and PF), while keeping the same degradation rates ($\gamma$). We again varied it from 3% to 30%, based on the maximum possible steady state of ODE oscillation (which is 1000).

## Testing the effect of different network motifs in the coupling degradation control to transcriptional oscillator

To test whether a certain network motif has better capability in controlling basal expression, we built partial models consisting of only the type-3 coherent feed-forward loop (CFFL), type-2 incoherent feed-forward loop (IFFL), negative feedback (NF), and positive feedback (PF) in either 'pure' transcriptional or combined transcriptional and degradation controls.

Here, we measured the concentration of the output gene (O) in the presence (I:ON, t = 40) or absence (I:OFF, t = 80) of an input signal (I), which we called the basal OFF condition. For the basal ON condition, we added the leakage term on both input and output genes (leakage level = 0.2, or 20% of the maximum steady state) and re-measured the concentration of the output genes. After that, we calculated the normalized difference of output gene expression,

which is defined as:

$$\text{normalized } \Delta O = \frac{(O_{basalON} - O_{basalOFF})}{0.2} \tag{48}$$

Finally, we subtracted the normalized $\Delta O$ of transcriptional model to the normalized $\Delta O$ of the combined transcription and post-translation model ($\Delta$normalized $\Delta O$) and report it as an indicator of how good a given motif is in controlling transcriptional leakage (Fig 4 and S4–S7 Figs).

## Supporting information

**S1 Fig. Sample trajectories of mRNA (left), protein (middle), and auto-correlation function (right) of one randomly chosen parameter set in model M1 (a) and M2 (b).**
(TIF)

**S2 Fig. The effect of transcriptional leakage on the robustness of an oscillator.** (Upper panels) box plot representing the normalized decay rates for each leakage level of model M1 (A), M3 (B), and M5 (C). (Lower panels) the percentage of parameter sets showing sustained oscillation under stochastic simulation. Red line indicates the median, and box edges indicate the 25th (Q1) and 75th (Q3) percentiles. The whiskers are defined as 1.5*(Q3-Q1).
(TIF)

**S3 Fig. Time trace trajectory of one randomly chosen parameter sets of (A) CFFL, (B) IFFL, (C) NF and (D) PF, with (lower panel) and without (upper panel) basal leakage.**
(TIF)

**S4 Fig. Contour plot representing the dynamics of type-3 coherent feed-forward loop in controlling transcriptional leakage under different parameter combinations.** I, E, and O representing the input, intermediate, and output genes, respectively. For simplicity, but without loss of generality, the production and the degradation rates of all genes was fixed at 1, the threshold value of I activation of E (K_Ei) at 0.5, and the Hill coefficient at 8. Furthermore, the leakage level (when Basal = ON) was fixed at 0.2 (or 20% of the maximum possible steady state).
(TIF)

**S5 Fig. Contour plots representing the dynamics of type-2 incoherent feed-forward loop in controlling transcriptional leakage under different parameter combinations.** I, E, and O represent the input, intermediate, and output genes, respectively. For simplicity, but without loss of generality, the production and the degradation rates of all genes was fixed at 1, the threshold value of I inhibition of E (K_Ei) at 0.5, and the Hill coefficient at 8. Furthermore, the leakage level (when Basal = ON) was fixed at 0.2 (or 20% of the maximum possible steady state).
(TIF)

**S6 Fig. Contour plots representing the dynamics of positive feedback in controlling transcriptional leakage under different parameter combinations.** I, E, and O represent the input, intermediate, and output genes, respectively. For simplicity, but without loss of generality, the production and the degradation rates of all genes was fixed at 1, the threshold value of O inhibition of E (K_Eo) at 0.5, and the Hill coefficient at 8. Furthermore, the leakage level (when Basal = ON) was fixed at 0.2 (or 20% of the maximum possible steady state).
(TIF)

**S7 Fig. Contour plots representing the dynamics of negative feedback in controlling transcriptional leakage under different parameter combinations.** I, E, and O represent the input, intermediate, and output genes, respectively. For simplicity, but without loss of generality, the production and the degradation rates of all genes was fixed at 1, the threshold value of O activation of E (K_Eo) at 0.5, and the Hill coefficient at 8. Furthermore, the leakage level (when Basal = ON) was fixed at 0.2 (or 20% of the maximum possible steady state).
(TIF)

**S8 Fig. Our general models showed similar dosage-dependent effect of proteasome degradation control on the robustness of the oscillator.** (Upper panel) box plot representing the normalized decay rates for each mutant condition of model M5 (A), and CFFL (B). Red line indicates the median, and box edges indicate the 25th (Q1) and 75th (Q3) percentiles. The whiskers are defined as $1.5^*(Q3-Q1)$. (Lower panel) The percentage of parameter sets showing sustained oscillation under stochastic simulation.
(TIF)

**S9 Fig. The distribution of mRNA (first panel), protein (second panel), Hill function of transcriptional control (third panel), and Hill function of degradation control (last panel) for one randomly chosen parameter set in M5.** The distribution was collected from the wild type (WT) (blue) or over-expression condition (red) with 10% basal leakage.
(TIF)

**S1 Appendix. Codes used in this study.**
(ZIP)

## Author Contributions

**Conceptualization:** Ignasius Joanito, Shu-Hsing Wu, Chao-Ping Hsu.

**Data curation:** Ignasius Joanito.

**Formal analysis:** Ignasius Joanito, Chao-Ping Hsu.

**Funding acquisition:** Chao-Ping Hsu.

**Investigation:** Ignasius Joanito.

**Methodology:** Ignasius Joanito, Ching-Cher Sanders Yan.

**Project administration:** Ignasius Joanito, Shu-Hsing Wu, Chao-Ping Hsu.

**Resources:** Ignasius Joanito, Shu-Hsing Wu, Chao-Ping Hsu.

**Software:** Ignasius Joanito, Ching-Cher Sanders Yan.

**Supervision:** Jhih-Wei Chu, Shu-Hsing Wu, Chao-Ping Hsu.

**Validation:** Ignasius Joanito, Chao-Ping Hsu.

**Visualization:** Ignasius Joanito.

**Writing – original draft:** Ignasius Joanito, Chao-Ping Hsu.

**Writing – review & editing:** Ignasius Joanito, Jhih-Wei Chu, Shu-Hsing Wu, Chao-Ping Hsu.

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
