## [Decision Letter · Decision Letter 0]

2 Apr 2020

Dear Dr. Hsu,

Thank you very much for submitting your manuscript "Basal leakage in oscillation: coupled transcriptional and translational control using feed-forward loops" for consideration at PLOS Computational Biology.

As with all papers reviewed by the journal, your manuscript was reviewed by members of the editorial board and by several independent reviewers. In light of the reviews (below this email), we would like to invite the resubmission of a significantly-revised version that takes into account the reviewers' comments.

We cannot make any decision about publication until we have seen the revised manuscript and your response to the reviewers' comments. Your revised manuscript is also likely to be sent to reviewers for further evaluation.

Sincerely,

Attila Csikász-Nagy

Associate Editor

PLOS Computational Biology

Jason Haugh

Deputy Editor

PLOS Computational Biology

Reviewer's Responses to Questions

**Comments to the Authors:**

Reviewer #1: Attached

Reviewer #2: The submitted manuscript explores the idea that post-translational controls can soften the negative impact of transcriptional leakage on oscillators. Transcriptional leakage is very common in promoters across different organisms, though it is known that basal production rates reduce the robustness of oscillators to parameter and molecular fluctuations. So when an oscillator is built upon transcriptional control, it is very reasonable to ask whether there is a mechanism in place to compensate for transcriptional leakage. The authors have identified two examples of this in the literature, one in the Arabidopsis circadian clock, and another in the mammalian clock.

I found the overall story to be interesting and a clear hypothesis has been tested.

*Introduction*

1. The authors state that the importance of post-translational mechanisms are rarely discussed in studies modelling the Arabidopsis clock. It would be better to substitute this subjective statement for a summary of the instances in which post-translational mechanisms *are* addressed in modelling work. For example, in Pokhilko et al., MSB 2012, the authors incorporate post-translational regulation of ELF3 protein by the E3 ligase COP1. This example is also highly relevant for the submitted manuscript.

2. Line 96-104. The observation that mutation of ZTL protein leads to a loss of oscillation period and amplitude robustness is interesting and deserves attention from modelling studies. It would be helpful to review existing theoretical analyses of the influence of degradation on oscillator robustness, beyond circadian oscillators, to help position the results of the submitted manuscript. Examples include:

a. Wong et al., MSB 2007: https://doi.org/10.1038/msb4100172

b. Krishna et al., PNAS 2006: https://doi.org/10.1073/pnas.0604085103

c. Caicedo-Casso et al., Sci Rep 2015: https://doi.org/10.1038/srep13161

After all, the results of Figures 1-4 are on abstract repressilator networks, as opposed to realistic models of circadian clocks.

*Results*

3. Line 133. The authors should include a bit more detail about the strategy for randomly generating parameter sets. What distributions are used for sampling parameter values, are how are those distributions chosen to be biologically meaningful? I can see from Table 1 that different ranges and scales are used, so I assume the sampling is done assuming uniform and log-uniform distributions. If so, add these details to the Methods section, and reference it accordingly in the Results.

4. In the first results section, the authors present results on what fraction of deterministic oscillators maintain their rhythmicity in the stochastic setting. I agree that this is interesting from the point of view of understanding how transcriptional noise might adversely affect the performance of an oscillator as a function of the type of inter-component control (transcriptional, post-translational, etc.). However, it does not incorporate any acknowledgement of the potential for noise to enable oscillations for parameter sets which do not oscillate deterministically, which is a well-studied phenomenon in ring oscillators. There is a nice paper on this from Strelkowa & Barahona (ICNAAM 2012, https://doi.org/10.1063/1.4756221), which additionally considers oscillation quality under different system sizes.

In light of this, I question whether it is relevant to ask what fraction of deterministic oscillators also oscillate stochastically. In subsequent results, how should one separate the effects of, for example, promoter leakage on deterministic oscillations and stochastic oscillations? Since stochastic oscillations is the setting of interest here, a better comparison of different architectures would be to simply compare what fraction of parameter sets oscillate stochastically, irrespective of deterministic oscillation.

5. Line 170. What is meant by “noise level” here? Figure 1D shows that the leakage level increases the normalized decay rate of the autocorrelation function. I don’t see why this is surprising. In the limit as promoters become inceasingly leaky, they become increasingly independent of their regulators, which can only result in unique stable equilibria, and no oscillations.

6. Line 206. The result presented in Figure 2 is interesting, but I wonder how robust it is to parameter uncertainty? In contrast to Figure 1, the results presented only show a single parameter set. By using distance measures for distributions (e.g. KL divergence), one could summarise the effect of leak itself as a distribution over different parameter sets, which would enable a more general statement to be made about the differences between M1 and M5.

7. Line 263. What does “analyzing it locally” mean in this context? I think this can be introduced more clearly. Furthermore, in this paragraph, it would be nice to point out that the parameter dependency of the conclusions has been analysed in the supplementary figures. This is easy to miss. For example, the text preceding the reference to Figure S3 doesn’t mention parameter variation.

8. Line 285. This section title is not clear. What observed behavior?

9. Line 294. What is the prevailing hypothesis in the literature for explaining how ZTL dosage influences oscillator robustness? The Somers paper doesn’t appear to mention transcriptional leakage, but maybe other studies have tried to explain this. It would be helpful to give greater coverage of the literature surrounding ZTL expression dosage.

*Methods*

10. Line 565: How the simulations were performed needs more detail. Which packages were used for ODE simulation? Or was custom code used? Which algorithms were used? Ideally, the code used to produce the results should be deposited in a public repository, for reproducibility. Perhaps the editor can comment on the PLoS CB policy on reproducibility and code depositing.

*Additional minor observations*

Author summary. 26s -> 26S.

Line 97: ZEITLUPE is spelled incorrectly.

Line 208: “when the mRNA transcribed is increased” -> “when the rate of mRNA transcription is increased”

Line 608: the reference to (96) doesn’t explain the decay of the autocorrelation function. That paper references a previous paper, which does explain it: d'Eysmond et al., Physical Biology 2013 (https://doi.org/10.1088/1478-3975/10/5/056005). Perhaps the d’Eysmond paper is a better citation to use.

Reviewer #3: Review of Basal leakage in oscillation; coupled transcriptional and translational control using feedforward loops

In this interesting paper, Joanito et al. explore in simple models the effect of transcriptional leakage on the robustness of oscillations. They provide computational evidence that post-translational control through targeted degradation can help buffer oscillators from the deleterious effects of transcriptional leakage. Although the results are interesting and are of general interest, there are a number of points that need addressing, mainly involving toning down some of the claims.

Major comments:

Novelty of results. The authors make a series of models based on the repressilator to generate their results. At a few points the authors suggest that post-translational control has been neglected in previous models of the Arabidopsis clock. The effects of post-translational control involving ZTL and COP1 have been included in previous Arabidopsis models (e.g PMID: 22395476, PMID:25033214). These models examine transcriptional leakage, but it is important to discuss previous examinations of post-translational control accurately, as I am not sure it is correct to say that these connections have been neglected. Additionally, previous work by this study’s authors in the plant circadian clock revealed that transcriptional feedforward loops can increase robustness (PMID:30224713) and in other clock systems feedforward loops have also been modelled (PMID:28007935). It is important that the authors discuss in greater detail how their new work relates to their previous work that found transcriptional feedforward loops improve robustness. They show that post-translational feedback loops are better than transcriptional ones in their model for robustness against transcriptional leakage, but it would be great to have further discussion of how post translational feed forward loops differ from transcriptional feedback loops.

2. Over statement of predictions from model. The authors claim that the greater variability observed in the ZTL mutant can be explained by the reduction in the level of posttranslational control. I am confused by these results. These data are from bulk averaging from many plants. Data at the single cell suggests that there is a great deal of variability between single cell clock rhythms (PMID: PMID:29697372). If the ZTL mutation really increased variability at the single cell level (which is presumably what their model is simulating), I would have thought that the rhythms would be damped out at the whole plant, or multiple plant level. The fact that plant single cell rhythms are variable, and that they are comparing a ‘single cell’ model of the plant to bulk averaged data should be discussed, especially given the work is examining noisy gene expression. In general, the predictions from the authors repressilator model should not be over empathised or linked to data too strongly. There are also newer papers examining the effects of ZTL on the plant circadian clock, beyond the Somers et al., 2004 paper. Do they show similar results?

3. Presentation: It would be excellent if more of the time traces of the simulated oscillations were presented in the main figures – to see more of the dynamics and to understand what the post-translational control is doing to the dynamics. Are their general effects that can be understood? Additionally, more examples could be shown in supplemental as well. It would help the reader understand the effects of post-translational control to observe more simulations directly.

4. Availability of models. Model code and data should be provided to allow readers to simulate models and evaluate themselves.

**Have all data underlying the figures and results presented in the manuscript been provided?**

Reviewer #1: Yes

Reviewer #2: No: Numerical results are not available in spreadsheets. For reproducibility of this purely computational study, it would be helpful to deposit code that reproduces the results.

Reviewer #3: No: It would be great if the data from the simulations and model code could were made available.

PLOS authors have the option to publish the peer review history of their article (what does this mean?). If published, this will include your full peer review and any attached files.

Reviewer #1: No

Reviewer #2: Yes: Neil Dalchau

Reviewer #3: No
---

## [Decision Letter · Decision Letter 1]

26 Jun 2020

Dear Dr. Hsu,

We are pleased to inform you that your manuscript 'Basal leakage in oscillation: coupled transcriptional and translational control using feed-forward loops' has been provisionally accepted for publication in PLOS Computational Biology.

Best regards,

Attila Csikász-Nagy

Associate Editor

PLOS Computational Biology

Jason Haugh

Deputy Editor

PLOS Computational Biology

Reviewer's Responses to Questions

**Comments to the Authors:**

Reviewer #1: The authors completely responds to the comments.

Reviewer #2: All of my comments have been addressed well in the revised manuscript.

Reviewer #3: The authors have addressed my concerns.

**Have all data underlying the figures and results presented in the manuscript been provided?**

Reviewer #1: Yes

Reviewer #2: Yes

Reviewer #3: None

PLOS authors have the option to publish the peer review history of their article (what does this mean?). If published, this will include your full peer review and any attached files.

Reviewer #1: No

Reviewer #2: **Yes: **Neil Dalchau

Reviewer #3: No

---

## [Editor Report · Acceptance letter]

14 Aug 2020

PCOMPBIOL-D-20-00268R1 

Basal leakage in oscillation: coupled transcriptional and translational control using feed-forward loops

Dear Dr Hsu,

I am pleased to inform you that your manuscript has been formally accepted for publication in PLOS Computational Biology. Your manuscript is now with our production department and you will be notified of the publication date in due course.

With kind regards,

Sarah Hammond
